# Simulation-guided engineering of split GFPs with efficient β-strand photodissociation

Yasmin Shamsudin [1,2] ✉, Alice R. Walker [1,3], Chey M. Jones [1], Todd J. Martínez [1] & Steven G. Boxer [1] ✉

Green fluorescent proteins (GFPs) are ubiquitous for protein tagging and live-cell imaging. Split-GFPs are widely used to study protein-protein interactions by fusing proteins of interest to split GFP fragments that create a fluorophore upon typically irreversible complementation. Thus, controlled dissociation of the fragments is desirable. Although we have found that split strands can be photodissociated, the quantum efficiency of light-induced photodissociation of split GFPs is low. Traditional protein engineering approaches to increase efficiency, including extensive mutagenesis and screening, have proved difficult to implement. To reduce the search space, key states in the dissociation process are modeled by combining classical and enhanced sampling molecular dynamics with QM/MM calculations, enabling the rational design and engineering of split GFPs with up to 20-fold faster photodissociation rates using non-intuitive amino acid changes. This demonstrates the feasibility of modeling complex molecular processes using state-of-the-art computational methods, and the potential of integrating computational methods to increase the success rate in protein engineering projects.

Green fluorescent proteins (GFPs) are the most widely used genetically encoded fluorescent reporters[1]. Since their discovery, GFPs have been the subject of exhaustive protein engineering efforts to enhance expression, stability, chromophore maturation rate, fluorescence quantum yield, color, and capacity for photoactivation, photoconversion, and photoswitching, the latter largely directed at applications for super-resolution imaging[1–4]. Split GFPs have been developed to probe protein-protein interactions by fusing fragments of the canonical GFP 11-stranded β-barrel to proteins whose interaction brings the fragments together, giving a fluorescence readout[1,5].

A shortcoming of split GFP complementation assays is that they are generally irreversible because the binding of the split β-strand peptide to re-form the intact, albeit still split, GFP is irreversible. While studying the properties of split GFPs, we were surprised to observe that once cut, some versions of split GFPs can be photodissociated[6,7], enabling optogenetic applications of GFPs along with their well-studied role for imaging. Photodissociation of the best-characterized example, a circular permutant of super-folder GFP with strand 10 at the

N-terminus and cut between strands 10 and 11, can be readily monitored by adding an excess of strand 10 containing the T203Y mutation that leads to a green-to-yellow color shift when it binds and replaces the photodissociated strand (Fig. 1a, b)[6,8]. Detailed investigation of this and other circular permutants led to the general potential energy surface (PES) for the photodissociation process shown in Fig. 1c[7]. Strand photodissociation was shown to be a two-step process in which light activates chromophore *cis-trans* isomerization, followed by light-independent strand-dissociation. Unfortunately, the quantum efficiency for this process is too low for practical applications[7,9]. Improving the efficiency of strand photodissociation, while at the same time preserving the stability of the split GFP against spontaneous thermal dissociation is a challenging undertaking given the complexity of the steps involved in strand photodissociation (Fig. 1c). Previous attempts using rational low-throughput approaches such as site-specific mutagenesis produced only modest improvements[7], while high-throughput methods using extensive mutagenesis and selection strategies proved very difficult to implement[10].

[1]Department of Chemistry, Stanford University, Stanford, CA 94305, USA. [2]Department of Chemistry-BMC, Uppsala University, 752 37 Uppsala, Sweden. [3]Department of Chemistry, Wayne State University, Detroit, MI, USA. ✉e-mail: yasmin.shamsudin@kemi.uu.se; sboxer@stanford.edu

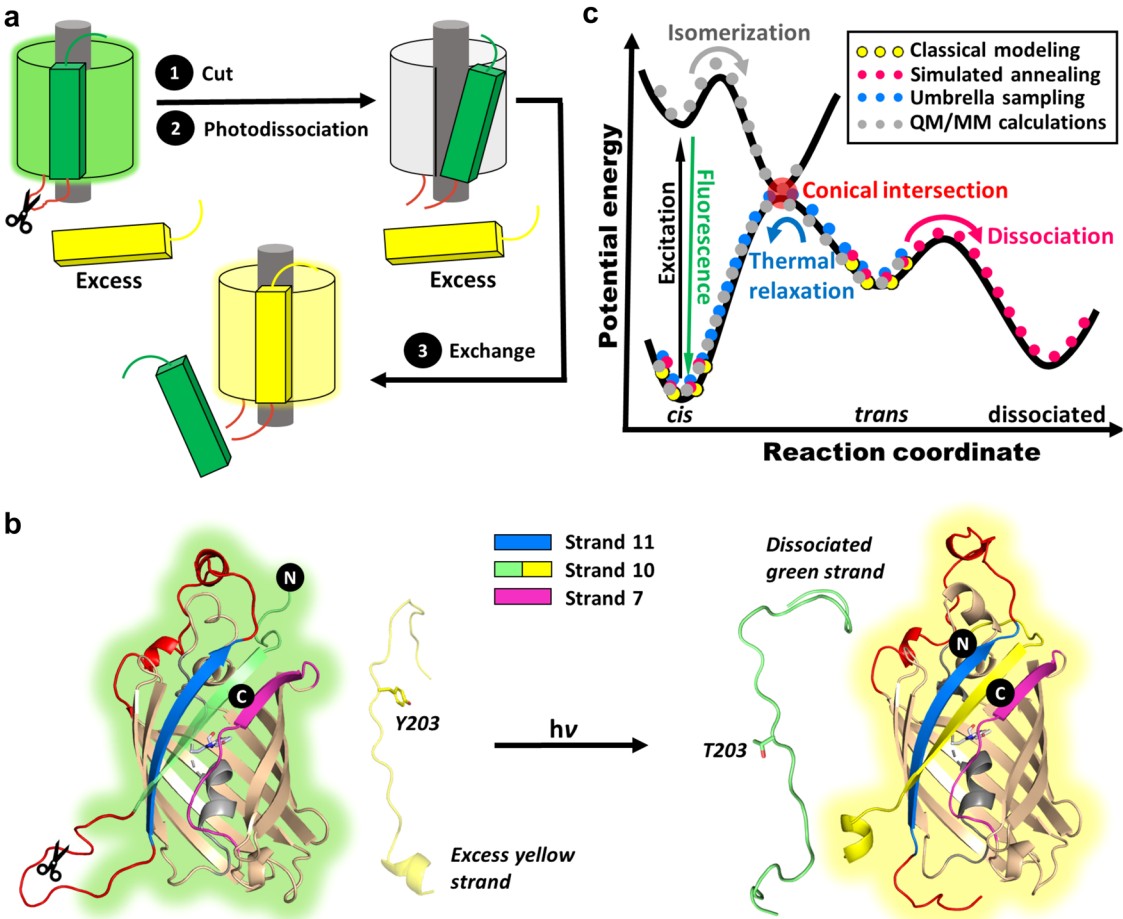

**Fig. 1 | Schematics of photodissociation. a** Scheme of photodissociation and strand-exchange experiments. The protein is cleaved between strand 10 and 11 and irradiated with a 488 nm laser, causing excitation, isomerization, and dissociation of the original strand 10 in the presence of excess synthetic strands containing T203Y. Upon binding, the synthetic strand shifts the absorption from green to yellow, enabling measurement of pseudo-first order exchange rates that reflect the rate of photodissociation. **b** Detailed view highlighting the N and C termini, internal alpha helix (gray), chromophore, the beta-strands 7 (purple), 10 (green/yellow), and 11 (blue), and the modeled loops on both ends of strand 11 (red). Scissors indicate the proteolytic cleavage site. **c** Schematic of potential energy curves for photodissociation (black lines[7]), overlaid with the computational methods used to model the states (dots). The excited and ground states are degenerate at the conical intersection (red).

As it is far from obvious what amino acid changes might enhance photodissociation while not adversely affecting spontaneous strand-dissociation in the dark, in the present work we aimed to limit the search space for mutagenesis by identifying key residues for mutation using computational modeling and simulations. Our simulations explore the structural landscape on the PES (Fig. 1c) to gain insight on the structural changes associated with the split protein during and in response to chromophore *cis-trans* isomerization. The simulation strategy described in the following leads to predictions of sites for mutagenesis (Fig. 2a) that were not expected by simple inspection of the crystal structure and guide the design of residues that are found to substantially increase the efficiency of photodissociation.

## Results

### Modelling and validating GFP structures

The computational procedure used to obtain models for predicting mutation sites for faster strand photodissociation and lowered isomerization barriers is outlined in Fig. 2a, with each step detailed in the Supplementary Information (S1–S8). A crystal structure of a circular permutant with strand 10 at the N-terminus is available (PDB entry 6OFO)[9]. However, as the structure contains two mutations (Cys48Ser and Cys70Ala), two unresolved loops, and multiple partially unresolved sidechains, mainly on the outside of the barrel, homology modelling

was required for further simulations. Using the crystal structure (Fig. 2b) as the template, three homology models (Fig. 2c) of the split-GFP complex with the chromophore (Fig. 2d) in the *cis* conformation (*cis* complex) and three homology models with the chromophore in the *trans* conformation (*trans* complex) were created. Each cleaved complex was subjected to three replicates of 1 μs unrestrained MD simulations to obtain consensus structures that could be used as starting structures for umbrella sampling and simulated annealing simulations. Upon visual inspection, we found that the barrels and associated sidechains were stable throughout the simulations, without showing signs of unfolding or spontaneous strand-dissociation. Furthermore, although the loops of the three homology models differed at the beginning of the simulations, they adopted similar conformations at the end. After 1 μs of simulations, the average heavy atom RMSD was 3.0 Å both for the nine simulations of the *cis* complexes, and for the corresponding nine simulations of the *trans* complexes. However, the average heavy atom RMSD of the peptide backbone in the β-barrel was <1.0 Å, with the main differences observed in the disordered loops (see Supplementary Fig. 1), indicating that all simulations had converged towards similar barrel structures. When comparing all 18 simulations, the average heavy atom RMSD was 3.6 Å. The average heavy atom RMSD of the peptide backbone in the β-barrel was still <1.0 Å, with the main differences in the loop regions, apart from the introduced changes

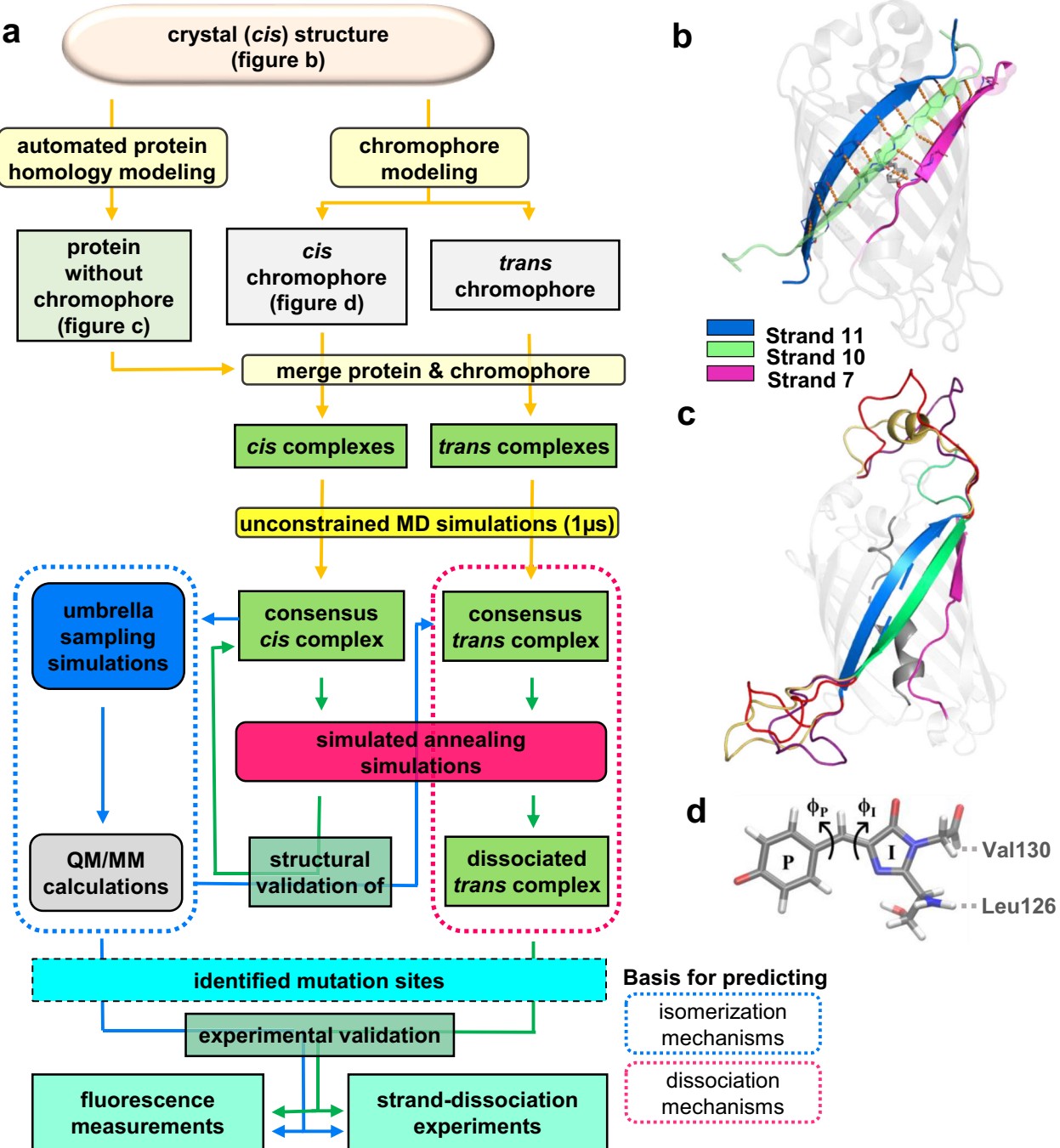

**Fig. 2 | Computational modeling and experimental validation. a** Computational protocol for generating models used for predicting dissociation (dotted magenta) and isomerization (dotted blue) mechanisms, leading to point mutations resulting in experimentally observed faster photodissociation or lowered isomerization barriers. **b** The crystal structure from the PDB entry 6OFO, highlighting inter-strand hydrogen bonds. **c** Homology models including loops (yellow, red, purple) missing in the crystal structure. The internal helix (gray) onto which the chromophores were modeled is highlighted. **d** Structure of the anionic chromophore moiety (*cis* configuration), illustrating the two possible twisting dihedral angles $\phi_P$ and $\phi_I$. Arrows illustrate clockwise rotation around the bonds. Dotted lines indicate capping atoms and neighboring residues. All protein images highlight the strands 7 (purple), 10 (green), and 11 (blue) for reference.

in the chromophore. Thus, one *cis* and one *trans* consensus complex was created using the final frame from one of the above trajectories of each chromophore conformation, and these *cis* and *trans* consensus complexes were then used as the basis for subsequent point mutations and enhanced sampling simulations.

The *cis* consensus complex was aligned with the template *cis* crystal structure, resulting in an average heavy atom RMSD of 1.0 Å, confirming a stable and conserved structure. Since no crystal structure exists for the corresponding *trans* complex, we used another

computational method to validate the *trans* consensus complex: the *cis* consensus complex was used as the starting point for classical umbrella sampling, gradually rotating and sampling the chromophore along its $\phi_I$ coordinate (Fig. 2d), thus modelling the *cis–trans* isomerization of the GFP chromophore in its protein environment albeit in the ground state. The heavy atom RMSD of the resulting *trans* complex at 160 degrees and the *trans* consensus complex was 2.0 Å, indicating high similarity despite different starting structures and computational approaches.

Apart from validating the *trans* consensus complex, the umbrella sampling simulations were used as starting structures for QM/MM calculations on the ground and excited states (see Supplementary Information S6). Although force field parameters such as atomic charges around the conical intersection cannot be well captured classically[11,12], our QM/MM calculations based on these classically sampled trajectories revealed that substantial differences in atomic charges were only found within a 10-degree window at the conical intersection ($\phi_I = 90-100°$) (Supplementary Fig. 2). Beyond this window, the umbrella sampling gave a plausible approximation for the description of the chromophore in the ground and excited states, as it moves from *cis* to *trans* and, therefore, the motion of protein residues around it. These results further validate the force field parameters used to simulate the *trans* consensus complex, while indicating the potential of analyzing the ground state umbrella sampling dynamics to assess how the protein environment changes around the chromophore as it isomerizes, which could guide the identification of mutation sites for decreasing rotational barriers for isomerization.

Finally, we assessed the stability of both the *cis* and *trans* consensus complexes through simulated annealing MD simulations (see Supplementary Information S8). Except for the rotation of the chromophore, only minor structural differences can be observed between the *cis* and *trans* complexes (overall average heteroatom RMSD: 3.6 Å). However, the simulated annealing simulations revealed how the differences in structure affect the stability of the split complexes. Notably, the barrel of the *cis* complex was intact even at high temperatures during the timescale of our simulations. By contrast, although the barrel in the *trans* complex was also very stable before heating, increased disorder, followed by strand-dissociation and unfolding, were observed during the high-temperature simulations (see discussion below). These results suggest that both consensus structures are stable for MD simulations at room temperatures. Experimentally, spontaneous strand-exchange in the *cis* form occurs very slowly over the course of weeks, but upon exposure to light, strand-exchange occurs in minutes to hours depending on incident power and the limited mutants that have been explored[7]. The observed different behaviors at higher simulated temperatures are consistent with these experimental observations, indicating that the models are useful for comparing the dynamics of both the *cis* and *trans* complexes over time.

## Simulating strand-dissociation and hydrogen bond analysis

The crystal structure of circularly permuted split GFP[9] reveals that the nearly ideal β-strand 10 is kept in place between the neighboring β-strands 7 and 11 (Fig. 2b). The main interactions between the β-strands in the barrel are inter-strand backbone hydrogen bonds formed between opposing main chain amides and carbonyl groups, especially between strands 10 and 11, and to a lesser degree, between strands 10 and 7. On strand 10, odd-numbered residues have their sidechains pointing into the barrel, while even-numbered residues have their sidechains pointing out of the barrel in the ground state structure (Fig. 3a). Besides the chromophore–sidechain interactions with Thr203 and Thr205 on strand 10 and His148 on strand 7, an inter-strand hydrogen bond involving a sidechain is seen only between Lys209 on strand 10 and the main chain of His217 on strand 11 (Fig. 3a). This observation was the basis of creating the Lys209Gln mutation to break the hydrogen bond, which resulted in up to two times faster photodissociation[7]. Unfortunately, no more obvious mutation sites can be inferred from the crystal structure, motivating the use of molecular modeling and simulations to obtain further structural information to guide additional mutations.

Through our combination of computational methods, we obtained models for studying the dynamic motions and structural rearrangements before, during and following isomerization, leading to disorder in the barrel and strand-dissociation. Simulated annealing is a well-established computational method for studying biophysical properties, including protein stability and conformational changes. Here, we use it to gain insight on how molecular interactions change as the strand dissociates from the β-barrel, to identify the main anchor points contributing to the stability of the strand-protein complex. The annealing protocol was carefully optimized to ensure that the dissociation process could be observed within a reasonable computational time frame. Interestingly, analyses of the simulated annealing simulations of the consensus *cis* complex revealed additional sidechain–sidechain interactions, mainly involving the outward-facing residues (Fig. 3b). On strand 10, Tyr200 stacks with Tyr151 on strand 7, while Ser202 alternatingly interacts to form hydrogen bonds with Asn225 on strand 11 and Asn149 on strand 7. Furthermore, Lys209 forms a salt-bridge with Asp216 on strand 11 while maintaining the hydrogen bond interaction with His217 discussed above. Interactions between sidechains on strands 7 and 11 and mainchains on strand 10 can also be observed (Fig. 3b).

To understand how isomerization affects these interactions, and the protein in general, we analyzed the structural changes and changes in hydrogen bond networks induced during the umbrella sampling simulations (Supplementary Fig. 3). Initially, as the chromophore isomerizes, the hydrogen bonds with Thr205 and His148 are broken. As the chromophore continues to rotate, interactions with the Asn121 sidechain and the Ser147 and Tyr151 mainchains are briefly observed before the chromophore reaches the *trans* state where it is partially solvated. Thus, mutation of Thr205, His148, and Asn121 could reduce steric hindrance along the isomerization pathway, resulting in the lowering of the rotational barrier, and thus an increase of the isomerization quantum yield. Whereas His148 maintains a hydrogen bond with Asn146 on the same strand, rotation of Thr205 results in the breaking of hydrogen bonds between the mainchain and the sidechain of Ser147 on strand 7. Several hydrogen bonds are broken and formed during the isomerization process. However, it is notable that several of the broken hydrogen bonds are between strand 10 residues and residues on strand 7, including between Thr203 and Ser147, but no new hydrogen bonds are formed between the two strands (Fig. 3c). This could explain the decreased stability of the *trans* complex compared with the *cis* complex, and the observed increased probability of strand-dissociation.

Finally, to elucidate the strand-dissociation pathway following isomerization, we analyzed the simulated annealing trajectories of the consensus *trans* complexes. At the start of these simulations, most inter-strand sidechain–sidechain and sidechain–mainchain interactions seen in Fig. 3a are broken. Particularly, the strand 11 sidechains Lys214, Arg215, and His217 near the cleavage site between strands 10 and 11 rapidly separate from strand 10; however, the complex remains stable until the subsequent separation of the Lys209 sidechain on strand 10 from both the His217 main chain and Asp216 side chain on the neighboring strand 11. The separation process follows the complete solvation of the cleaved end of strand 10 (Fig. 3c) and is only observed during heating and simulations at 600 K (see Supplementary Information S8). Once Lys209 separates, the complex becomes increasingly disordered, especially around strands 10 and 7, and at the cleaved end of strand 11 (Fig. 3c). Notably, before the strand becomes completely disordered and detached, we can still observe the alternating sidechain–sidechain hydrogen bond interactions between Ser202 and the neighboring Asn225 and Asn149. Meanwhile, Tyr200 loses the stacking interaction with Tyr151, and new hydrogen bond interactions are formed with Asn149. At this point, mainchain–mainchain interactions can still be seen around Tyr200 and Thr205 (Fig. 3c). Although the preceding mechanisms and structures can be observed in all replicates of the high-temperature simulations of the *trans* complex, different scenarios emerge as the simulations progress from the structures equivalent to that illustrated in Fig. 3c. In some simulations, the gap between strands 10 and 7

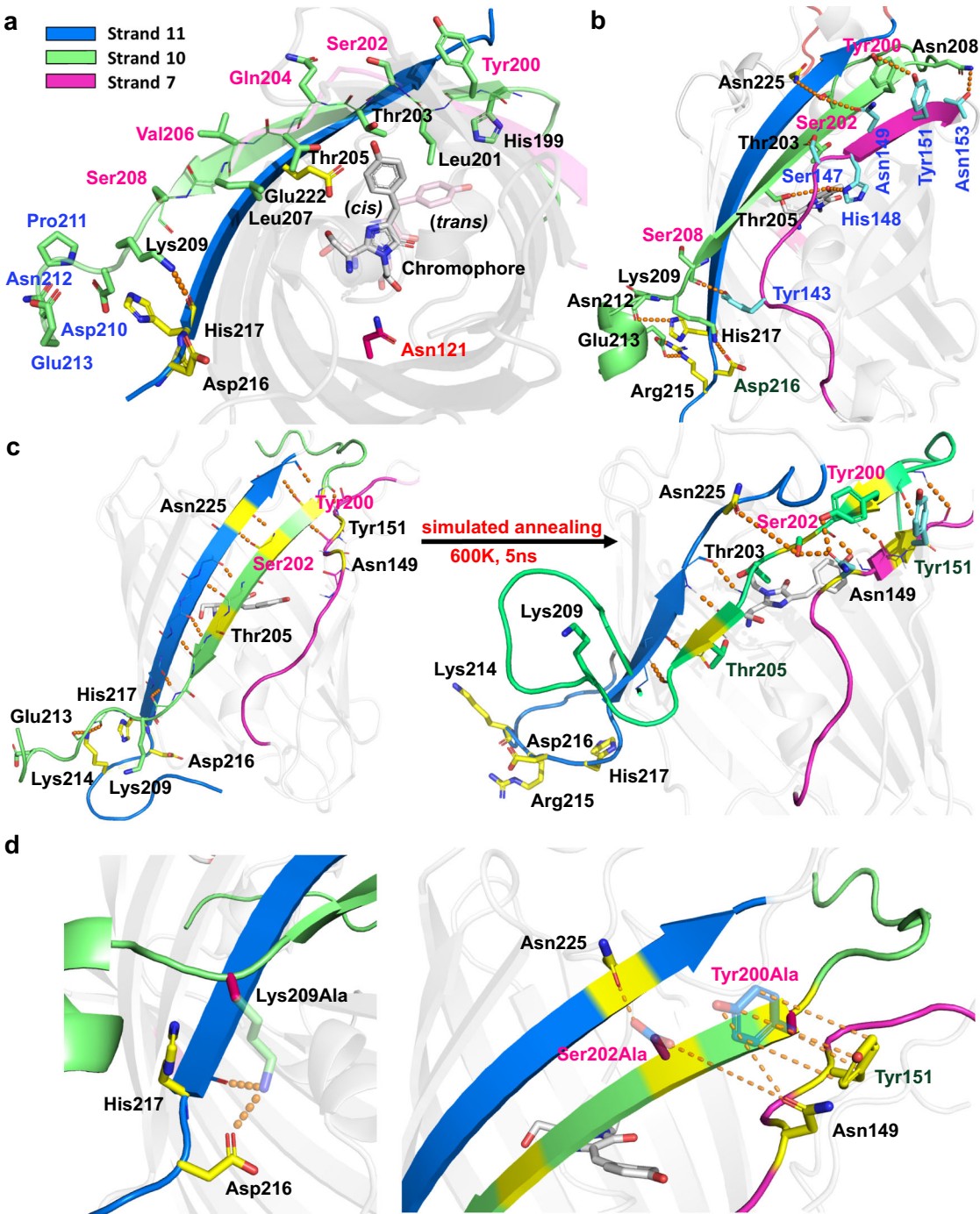

**Fig. 3 | Detailed view of inter-strand interactions.** Overview of the hydrogen bonding (orange dashes) network between strand 10 (green) and the neighboring strands 7 (purple) and 11 (blue) in the (**a**) consensus *cis* complex highlighting the inter-strand hydrogen bond between Lys209 and His217. Sidechains on strand 10 color-coded as follow: pointing out of the barrel (pink), into the barrel (black) or fully solvent-exposed (blue). Glu222 (yellow) and Asn121 (magenta) shown for reference. **b** *Cis* complex following simulated annealing highlighting inter-strand side-chain interactions. **c** Consensus *trans* complex illustrating the separation of Lys209 from His217 and Asp216, and the fully solvated cleaved end of the strand 10 before heating, and the partially disordered state where inter-strand sidechain-sidechain interactions involving strand 10-residues have been reduced to only Tyr200 and Ser202 after heating. Labeled residues (yellow) shown for reference, and (**d**) our identified mutation sites Lys209, Ser202, and Tyr200. Transparent sidechains show original amino acids, while dashes indicate removed polar inter-actions following mutations to alanine.

increases while mainchain–mainchain interactions between strands 10 and 11 remain, ahead of strand-dissociation. In others, the strand 10 moves in between strand 7 and 11, shifting the positions of strand 10 residues relative to the neighboring strands in a stepwise manner. Thus, as Asn225 and Asn149 lose their interactions with Ser202, these are replaced by the corresponding interactions with the next outward-facing residue on strand 10, Tyr200.

## Protein engineering predictions

As shown schematically in Fig. 1a, photodissociation of split strand 10 can be readily measured by irradiating the split-GFP complex in the presence of an excess of a strand 10 peptide containing the T203Y mutation that converts the protein with an absorption at 470 nm (GFP) into one that absorbs at 505 nm (YFP-like). To simplify comparison of strand-exchange rates and validate the predictability of our models, we

introduced three criteria for our protein design aimed at faster strand photodissociation. First, mutations are only introduced on the dissociating strand so that identical products result when the excess peptide binds to form YFP. This ensures that we measure the formation rate of identical YFP products, simplifying comparisons of GFP mutants. Second, as changes in the environment around the chromophore could lead to confounding changes, e.g., spectral changes, isomerization behavior, and chromophore maturation rates, our second criteria was to minimize interference with the chromophore on the interior of the barrel by considering only solvent-exposed residues (see Fig. 3a). Although counterintuitive, the simulations suggest that such residues could be important along the dissociation pathway. Third, to separate and assess the reliability and predictiveness of our models for isomerization and strand photodissociation, single-point mutations should be identified as either affecting isomerization or strand photodissociation, but not both.

To identify point mutations that could affect dissociation separately from isomerization, Fig. 3b highlights several interactions between strand 10 and the neighboring strands. Dissociation occurs in the *trans* complex, shown in Fig. 3c, and here the number of interactions between strand 7 and neighboring strands has been drastically reduced, significantly decreasing the number of potential mutation sites. Our simulations indicate that the hydrogen bond between Lys209 and His217, and the salt bridge between Lys209 and Asp216 are stable at 300 K. Since the separation of Lys209 from both precedes the disordering of strand 10, mutation of Lys209 could enhance photodissociation rates; however, the single-point Lys209Gln mutation resulted in only modest improvements[9]. Although Gln209 would not form a salt-bridge with Asp216, it can still form a hydrogen bond, thus likely limiting the effect of the mutation. Meanwhile, a non-polar residue, such as isoleucine, would prevent such interactions, while an alanine would also reduce non-polar sidechain interactions. Note that residue 209 is the last residue at the end of β-strand 10 and so it is far from the chromophore.

Apart from Lys209, we identified Ser202 and Tyr200 as two additional candidates for mutation. In the simulated annealing simulations, Ser202 forms alternating hydrogen bonds with Asn225 on strand 11 and Asn149 on strand 7. Meanwhile, Tyr200 is either stacked with Tyr151 on strand 7 or hydrogen bonded to Asn149. Although these interactions are also present in the *cis* complex, the *cis* complex has additional inter-strand interactions (Figs. 2b and 3b) that stabilize the protein and could reduce the effects of the mutations at these sites. Since the *trans* complex has fewer stable inter-strand interactions (Fig. 3c), the relative effect of these mutations on strand-dissociation should be larger. Thus, while Tyr200, Ser202, and Lys209 could be independently mutated, our simulations suggest that a combination of mutations of all three would produce the largest impact on strand-dissociation (Fig. 3d). To reduce their polar sidechain interactions, we considered two alternatives—eliminating sidechain interactions through alanine mutations, and introducing steric clashes using bulky and nonpolar residues, such as isoleucine, valine, leucine, and tryptophan (Fig. 4a). Note that traditional alanine scans change one residue at a time, and it would be very time consuming to scan all permutations of three alanines.

Although improving the efficiency of strand-dissociation is the focus of this work, it could also be desirable to introduce mutations that reduce the isomerization barrier to increase the yield of the dissociating *trans* complex (Fig. 1c). The umbrella sampling simulations and the subsequent hydrogen bond analysis indicate three sites where mutations could affect isomerization: Thr205, His148, and Asn121. Thr205 on strand 10 and His148 on the neighboring strand 7 could affect both isomerization and strand-dissociation as isomerization precedes strand-dissociation, so it would be difficult to untangle the contribution to each step at such mutation sites. Meanwhile, Asn121 is situated on the side of the chromophore opposite the dissociating strand (Fig. 3a), making it a suitable candidate for investigating the potential impact on isomerization separately from strand-dissociation.

## Strand-dissociation experiments test predictions

As Tyr (Y) 200, Ser (S) 202, and Lys (K) 209 are not directly interacting with the chromophore, and are not expected to directly affect isomerization, changes in kinetics resulting from mutations at these sites are expected to mainly affect strand-dissociation. Henceforth, our reference structure is referred to as YSK (the mutations are listed by the amino acids at the mutated positions from the N- to C-termini, Fig. 4a). To test our predictions, four single-point mutants, one double mutant, and six triple mutants (Fig. 4b), as well as the reference structure were expressed, cleaved, and subjected to laser-induced strand-exchange experiments (Fig. 1a, Supplementary Information S9–S18). Our results show that among the three mutation sites, the Ser202Ala mutation caused the largest improvement for the single-point mutants. Additionally, the inclusion of the Ser202Ala mutation in double and triple mutants consistently improved strand-exchange rates. Furthermore, all our triple-mutants display faster rates compared with our reference protein YSK, indicating the usefulness of our models. Most notably, the AAA mutant displays the largest rate increase, up to 20 times faster than YSK (Fig. 4b, Supplementary Table 1). To ensure that the mutation did not also equivalently increase spontaneous dissociation in the *cis* form, we also monitored strand-exchange rates of AAA samples not exposed to light (dark controls). The dark exchange rate was 85 times slower than the light-induced exchange rate.

## Fluorescence quantum yield experiments and photoisomerization

Fluorescence quantum yields (FQYs) of GFP mutants have previously been measured to estimate *cis*–*trans* photoisomerization efficiency. Although changes in fluorescence are not necessarily correlated with photoisomerization, FQYs can nevertheless be used to probe the excited-state behavior of mutants, as excited-state isomerization competes with fluorescence emission. For the triple mutants AAA and IWI, we observed 2-4% increases in FQY (Supplementary Table 2), suggesting that the point mutations on the dissociating strand do not have a substantial effect on the photoisomerization efficiency. Thus, we can deduce that changes in observed exchange rates in our single, double, and triple mutants are primarily correlated to the strand-dissociation rates.

To assess our model of the isomerization pathway, two quadruple mutants were expressed, based on the AAA and IWI mutants, adding the Asn121Ala mutation, and referred to as AAA-A and IWI-A; however, only IWI-A yielded enough protein for further experiments. To investigate if the Asn121Ala mutation affects the isomerization barrier, we measured its FQY. Interestingly, the experiments revealed a decrease by 8% compared with the corresponding IWI mutant and a 4% decrease compared with the YSK reference structure. Although this result does not prove that the mutation increases the *cis*–*trans* photoisomerization efficiency, it could be one explanation for the results. To validate this assumption, we performed umbrella sampling simulations of the IWI-A mutant, which showed lowered isomerization barriers of 5 kcal/mol compared with YSK (see Supplementary Fig. 4), indicating that increased *cis*–*trans* isomerization is at least partially the reason for the decreased FQY. Interestingly, despite this increase in the population of the *trans* complex, the strand photodissociation of IWI-A did not increase relative to IWI, and the exchange rate for IWI-A decreased compared with IWI.

## Discussion

β-strand photodissociation of split GFP is a complex process, and conventional directed evolution methods to increase the photodissociation yield while not enhancing spontaneous dissociation in the dark have

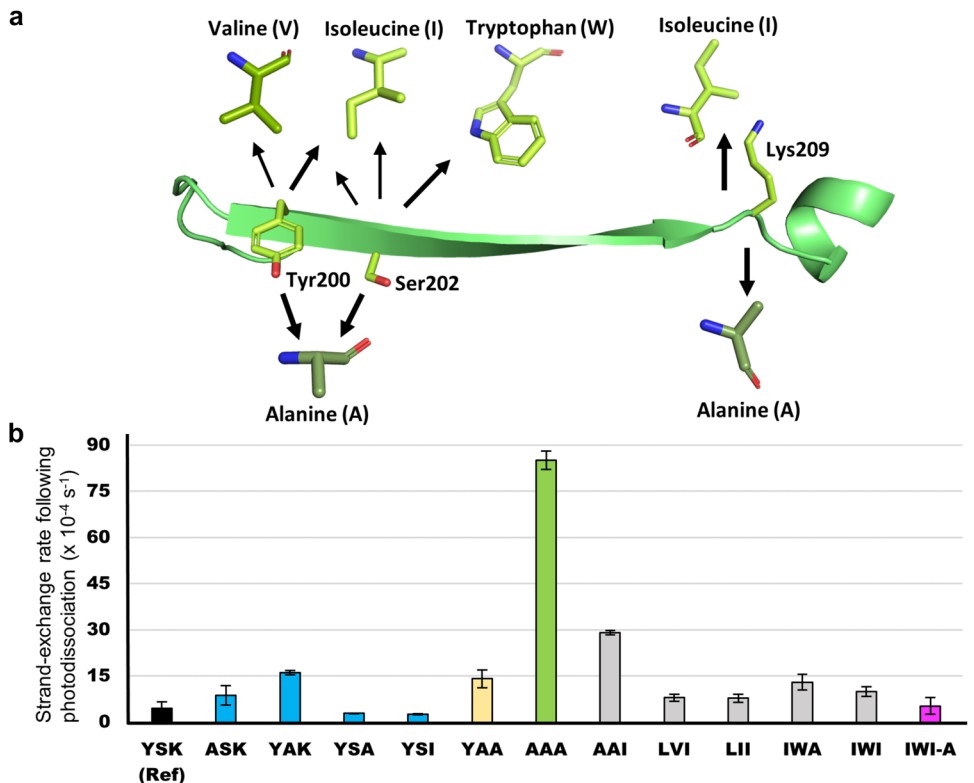

**Fig. 4 | Engineered mutants and their strand-exchange rates. a** Mutant combinations considered in this work. Our reference structure containing Tyr200 (Y), Ser202 (S), and Lys209 (K) is referred to as YSK and shown on the strand using red labels. The mutants are denoted by the amino acids in these positions and orders. **b** Pseudo-first order rates of strand-exchange (Fig. 1a) following photodissociation for the YSK (black) and our mutants. Data are presented as mean values +/- SD between $n = 3$ independent experiments. Single-point mutants are highlighted in blue, the double mutant in yellow, triple mutants in grey, with the fastest photo-dissociating triple mutant AAA highlighted in green, and the quadruple mutant in pink, which includes the IWI mutations on the strand, and the additional Asn121Ala mutation. The spontaneous strand off rate in the dark is in the baseline. 488 nm laser irradiation at 26 mW was used for all experiments. The source data are provided as a Source Data file.

proven difficult[10]. Therefore, we turned to simulations, even though the overall process being simulated requires multiple approaches, including multiscale modeling and enhanced sampling methods, and pushes the limits of what is possible. Based on molecular modelling and simulations, we have made predictions and then engineered a series of GFP mutants with specific properties, namely split GFPs with substantially more efficient β–strand-dissociation following photoexcitation.

Here, various computational methods were used to model the entire pathway of *cis*–*trans* isomerization and the subsequent strand-dissociation process. Although there are clear limits to the usefulness of the methods used here, there are also obvious advantages. For instance, the complexity of the excited-state isomerization makes simulations in the ground state at best an indirect approach to a true excited-state calculation; however, the latter are computationally very expensive, especially if coupled to simulations of the protein dynamical response. Meanwhile, although the protein movement caused by the chromophore isomerization in the excited state can be approximated by simulations in the ground state, the corresponding calculated barriers of isomerization in the ground state are likely not accurate. The goal here, however, was to visualize the trajectory of strand-dissociation, inspect the structures for interactions that are broken during strand-dissociation, and then make and test predictions based on this model. Similarly, although the simulated annealing simulations proved to be a useful method for visualizing the dissociation mechanism and identifying key residues for point mutations, using such simulations to predict how much faster a mutant would dissociate compared to the reference structure is challenging.

With these results in hand, it could be posited that traditional protein engineering methods such as alanine scanning would have arrived at the same results, without the need for computational modeling. However, even an alanine scan limited to the residues pointing out of the barrel and the solvent exposed ones at the cleaved loop site would be 9 single-point mutations, and additional 2 and 3 random combinations of these would be an enormous undertaking as strand exchange experiments are time-consuming. If we only consider alanine mutations, our computational method resulted in 6 mutants, including the double and triple mutants. Additionally, we note that the Lys209Ala single-point mutation did not have improved dissociation rates compared with the reference structure, although it is a precursor to the fast-dissociating AAA mutant. Thus, it could have been discarded in a traditional workflow, and the subsequent triple mutant would not have been created.

Nonetheless, through the combination of theoretical, computational, and experimental methods, we have demonstrated an approach to engineering and modifying a complex protein function, producing what could be a form of split GFP that should be useful both as an optogenetic and imaging tool. The ability to efficiently dissociate the strand with light opens possibilities of site-specific reversible macromolecular interactions, genetically encoded caged enzymes, and site-specific cargo delivery in cells[13].

## Reporting summary

Further information on research design is available in the Nature Portfolio Reporting Summary linked to this article.

## Data availability

The data that support this study are available from the corresponding authors upon request. The structure 6OFO was used in this study and is

accessible from the Protein Data Bank (PDB). Details on computational and experimental methods, including protein and chromophore modeling, MD and QM/MM simulations methods, protein expression and purification, mass spectrometry, absorbance measurements, dissociation experiments, and fluorescence measurements, as well as extended discussions are available in the Supplementary Information[14–31]. Files associated with performed simulations, including starting structures, parameters, and short videos of simulation trajectories, and source data underlying Fig. 4b and Supplementary Tables S1, S2, and S4 can be retrieved from the Zenodo server [https://doi.org/10.5281/zenodo.7674800]. Source data are provided with this paper.

## Code availability
Simulation input files can be retrieved from the Zenodo server using the following URL: https://doi.org/10.5281/zenodo.7674800.

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

## Acknowledgements
We thank the Stanford Research Computing Center for the computer resources used in this project. We thank Drs. Chi-Yun Lin and Matt G. Romei for valuable discussions. We thank Dr. Chu Zheng for helping us with expressing, mutating, and purifying proteins. We also thank Jacob Kirsh for assistance with protein purification and for valuable discussions. Y.S. was supported by the Knut and Alice Wallenberg Foundation. This work was supported in part by NIH Grant GM118044 (to S.G.B.) and by the U. S. Department of Energy, Office of Science, Office of Advanced Scientific Computing Research, Scientific Discovery through Advanced Computing (SciDAC) program and the Chemical Sciences, Geosciences, and Biosciences Division of the Office of Basic Energy Sciences, Office of Science, U. S. Department of Energy, AMOS program (to T.J.M.).

## Author contributions
Y.S. designed the study, modeled the chromophore and protein complexes, performed classical and annealing simulations, and the experiments. A.R.W., and C.M.J. performed the umbrella sampling and quantum chemical validation. Y.S., A.R.W., and C.M.J. wrote the manuscript. Y.S., A.R.W., C.M.J., T.J.M., and S.G.B. edited and revised the manuscript.

## Funding

## Competing interests

The authors declare no competing interests.
