## [Peer Review File · Nature Communications]

Simulation-guided engineering of split GFPs with efficient β -strand photodissociationReviewers' Comments:

Reviewer #1:

Remarks to the Author:

This manuscript provides a nice case of using molecular dynamics simulation to assist protein engineering. The main merit of this manuscript is the difficulty of this target system for both traditional mutation-selection approach because of the lack of a high-throughput selection method, and for computational simulation, because of the involvement of photophysics. QM/MM simulation in this manuscript has, to some extent, reduced the search space for the engineering effort, resulting in a split GFP with much improved photodissociation rate. On the other hand, the resulted improvement still seems to be still insufficient for practical applications. It would be helpful for the manuscript describe the dissociation rate needed for the practical use of the photodissociation system. Overall, this manuscript illustrates the promise of computational simulation in protein engineering involving photophysics and photochemistry, although the practical advancement achieved is very limited.

Reviewer #2:

Remarks to the Author:

The authors employ state-of-the-art computational methods (by combining classical and enhanced sampling molecular dynamics with QM/MM calculations) for modeling complex molecular processes such as the key states involved in the protein-protein photodissociation process of split-GFPs (that have been developed to probe protein-protein interactions by fusing fragments of the canonical GFP to proteins whose interaction brings the fragments together, giving a fluorescence readout). Strand photodissociation was shown to be a two-step process with too low quantum efficiency to be relevant for practical applications. A novel computational approach has been proposed here as an alternative to the standard extensive mutagenesis and selection strategies (e.g., inspection of the crystal structure to guide the design) that have been so far unsuccessful in the design. This approach and its predictions, that have been experimentally tested and validated, eventually enables the rational design and engineering of split GFPs (involving non conventional amino acids) with up to 20-fold faster photodissociation rate, thus paving the way to novel optogenetic applications of split-GFPs (along with their well-studied role for imaging). Very notably, the most effective fast-dissociating AAA mutant would have been overlooked by employing a traditional workflow.

I believe this is a great piece of work, very clearly written and accessible to a more general non-specialized audience, showing a novel computational approach to protein engineering that has been experimentally validated.

It is my opinion that this works deserves fast publication on NatureComm as it is.

Reviewer #3:

Remarks to the Author:

In this sweet and short paper that I enjoyed reading, the authors, provide a multi-scaling theoretical framework (molecular dynamics combined with hybrid quantum mechanics and molecular mechanics approach) that enables rational design of split GFP with dozen-fold faster photodissociation rates using non-intuitive amino acid point mutations. What I like is the general applicability of the proposed framework for modeling of complex and large systems with a affordable computational costs. The work shows significant progress in the development of a useful computational toolkit that have been applied to investigate the entire pathway of cis-to-trans of GFP isomerization and the subsequent strand-dissociation process.

The paper is very well-written, the way how the simulations were performed is clearly described,

which guarantees the reproducibility. The conclusions drawn – together with taking into account existing literature – are well-founded. The results obtained by the candidate are important for advancing the field of protein engineering. In particular, the developed methodology would have a potential impact on answering fundamental questions in optogenetics and photo-pharmacology. The work highlights the interplay between the experimental and theoretical studies and illustrating the power of such a combined effort, with an emphasis of the role of theory and computation leading experimental studies in the search for new light-activated molecules that open possibilities for site-specific drug delivery and controlling biological process.

I typically do not accept papers as it is, however, there is nothing I can think of that can further improve the manuscript.

We thank the referees for reviewing our work and for the comments.